# Phylogenetic Aspects of Higher Plant Lipid Fatty Acid Profile

**DOI:** 10.3390/ijms26199424

**Published:** 2025-09-26

**Authors:** Alexander Voronkov, Tatiana Ivanova

**Affiliations:** K. A. Timiryazev Institute of Plant Physiology, Russian Academy of Sciences, 35 Botanicheskaya St., 127276 Moscow, Russia; itv_2006@mail.ru

**Keywords:** acyl chain length, average chain length, even-chain fatty acid, odd-chain fatty acid, polyunsaturated fatty acids, temperature of phase transition, unsaturation index

## Abstract

Humans have been using lipids for many centuries; these are oils found in plants, particularly in seeds. However, relatively recently, it has become clear that lipids are the primary metabolites of any living organism. Fatty acids (FAs) are a structural component of lipids, and their role in building the framework of the lipid bilayer cannot be overstated. They participate in maintaining homeostasis by controlling membrane permeability. Changes in the FA composition of lipid bilayers can modulate the transition of the membrane from a liquid crystalline to a gel-like state. Thus, knowledge of a plant’s FA profile can aid in understanding the physiological mechanisms underlying their interaction with the environment and the ways in which they adapt to various stress factors. Throughout the colonization of terrestrial habitats, plants evolved, and new phylogenetic groups appeared; at present, some features of the FA composition of their individual representatives are known. However, the overall change in the composition of lipid FAs during the evolution of higher plants is still not understood. Our analysis of the literature showed that the FA diversity tends to decrease from mosses to angiosperms, mainly due to a reduction in polyunsaturated very-long-chain FAs, while the average acyl chain length remains unchanged. It is important to recognize the trends in this process in order to understand the adaptive capabilities of higher plants. This knowledge can be useful not only from a fundamental point of view, but also in practical human activities.

## 1. Introduction

Only a little over 70 years has passed since Watson and Crick discovered the structure of the DNA molecule in 1953 [1]. However, considerable progress has now been made in understanding the structure of genes and the proteins they encode, and many articles and reviews have been devoted to their evolutionary transformations [2,3,4,5,6,7,8]. Despite the chemical structure of lipids and their constituent fatty acids (FAs) being known for over three centuries, our understanding of how they are synthesized and their evolution in different plant groups is still incomplete [9]. While one can find information in the literature about the evolution of FA desaturases—proteins involved in FA synthesis—there is practically no information about changes in the composition of lipid FAs during the evolution of plant organisms [10,11]. Therefore, in this review, we summarize information on more than 550 higher plants (Embryophyta), including mosses (Marchantiopsida, Jungermanniopsida, Sphagnopsida, Bryopsida, and Polytrichopsida), ferns (Polypodiopsida), gymnosperms (Gnetopsida, Ginkgoopsida, Cycadopsida, and Pinopsida) and angiosperms (Magnoliopsida) (Table A1).

It should be noted that the analysis of literary sources covers a period of more than half a century. All this time, gas chromatography (GC) has been and remains the gold standard for analyzing the FA composition of lipids [12,13]. Only detection methods have undergone a significant change, with gas–liquid chromatography coupled with mass spectrometry leading to the considerable enhancement of analysis and the detection of many rare FAs that were previously only detectable through targeted analysis using specific standards [14,15]. However, the reliability of the GC method enables a comparison of results from different years, as minor FAs rarely account for more than a one percent, and the main composition of FAs is not in doubt. In addition, it is obvious that different plant organs performing different functions may feature in the composition of FAs [16,17]. At the same time, during the evolutionary process, the number of organs and tissues increased, which makes it more difficulty to perform a comparative analysis. Therefore, we used literature sources that mainly studied the composition of FAs in the photosynthetic part of plants. This helped us avoid potential errors in data interpretation (compared to various storage organs (fruits and seeds) or non-photosynthetic organs (for example roots)) and enables us to trace the reasons for phylogenetic changes in the FA composition of Embryophyta.

## 2. Diversity of Fatty Acids

The monocarboxylic acids comprising the lipids of higher plants are represented by 144 individual FAs (iFAs) (Appendix A). The iFAs number in individual plant species ranges from 2 to 69 (Figure 1). The greatest variation is observed in mosses, but this is slightly lower in ferns (from 10 to 65) and lowest in gymnosperms (from 14 to 25) and in angiosperms (from 2 to 30). Generally, (median value), there are 28 iFAs in mosses, 27 in ferns, 20 in gymnosperms, and 12 iFAs in angiosperms. Thus, during the evolution of Embryophyta, there is a clear reduction in the composition diversity of iFA lipids.

There are several theories that explain the reasons for this trend. First, mosses, as non-vascular plants, are highly dependent on environmental humidity and are prone to drying out. Their complex lipid composition may play a role in maintaining membrane fluidity and stabilizing it during drying/rehydration cycles. Vascular plants, especially angiosperms, have developed conducting systems, cuticle, stomata and other adaptations that allow them to better control their water balance, reducing the selective pressure required to maintain an extremely complex lipid composition to protect membranes from dehydration [18]. Secondly, as plants evolved and adapted to diverse terrestrial conditions, lipid metabolism became optimized and specialized. Angiosperms developed more efficient and ‘targeted’ lipid synthesis pathways, focusing on the key FA necessary for their complex structures (e.g., developed photosynthetic apparatus, flowers, fruits) and functions [19]. This led to the loss of ‘excessive’ or less critical pathways for the synthesis of exotic FAs, characteristic of more ancient groups. It is known that many angiosperms have lost the ability to synthesize 16:3(7,10,13) in chloroplasts–plastidial or prokaryotic pathway [20,21], and that they have switched to predominantly using the extra-plastidial or eukaryotic pathway (synthesis of 18:1(9) in plastids followed by desaturation in the endoplasmic reticulum) [22,23]; this has undoubtedly led to a reduction in the diversity of the FA profile. Thirdly, it is known that many unique and rare FA mosses can perform specific protective functions (antimicrobial, antifeedant, cryoprotective) in their relatively primitive and vulnerable structures [24,25]. The most striking example of such FAs is acetylenic acids, which are often found in mosses and practically never in other tribes (Appendix A); they can therefore be used as marker acids for mosses (Bryophytes) [26]. Acetylenic FAs are credited with powerful antifungal properties [27] and protect plants against being eaten [28]. In angiosperms, these functions could have been partially taken over by other classes of compounds (e.g., flavonoids, alkaloids, terpenoids, complex waxes), thus reducing the need for the synthesis of a wide range of specialized FAs [29,30].

## 3. Major Fatty Acids

In the vast majority of Embryophyta species, the major FAs are 16:0, 18:2(9,12), and 18:3(9,12,15) (Appendix A). In angiosperms, with the exception of Nicotiana [31], the major FAs also include 18:0 and 18:1(9). In addition to the above, the major FAs in gymnosperms include 20:0 and 20:3(5,11,14). In ferns and mosses, the major FAs also include 18:1(9) and 20:4(5,8,11,14) (Appendix A). At the same time, the total amount of major FAs in different species ranges from 33% to almost 100%, averaging 85% (median 86%) of all FAs.

The relative uniformity of the major Embryophyta FAs can be explained by the energetic optimization of the pathways of their synthesis. The main pathway of FA biosynthesis in plants is highly conserved, and its enzymes, such as β-ketoacyl-ACP synthases, are optimally adapted to work with precursors of a certain chain length (mainly 16C and 18C). The desaturase enzymes responsible for the double-bond formation (such as FAD2, FAD3, FAD7, FAD8) are also highly conserved and specific to 18C substrates [32]. Thus, the production of longer or more unsaturated FAs requires additional enzymatic steps and the expenditure of ATP and NADPH. In addition it can be assumed that evolution did not “reinvent” the entire metabolic pathway for major FAs, since the existing one perfectly meets the basic needs of plats.

However, it should be noted that there are many variations in the major FAs of different species. For example, in the aforementioned Nicotiana, the major ones are 16:1(9) and 16:3(7,10,13) (Appendix A). There are species in which 12:0 and 14:0 can be classified as the major FAs (Appendix A). It is likely that such specific differences between certain species are related to the peculiarities of their biology. For example, in the Solanaceae family, significant amounts of FA have been found in secreted sugar polyesters, which are associated with the broad protective properties of the secretion against pathogens and insect herbivores [33,34,35,36,37].

## 4. Acyl Chain Length

FAs contain varying numbers of carbon atoms in the acyl chain. Based on this characteristic, all FAs are divided into groups, but it should be noted that there is no single established classification, and these groups of FAs vary greatly between plant and animal objects. During our analysis of the literature, we found that the following are most often considered for plant objects: short-chain: ≤6C [38,39,40], medium-chain: 7C–13C [38,41,42,43,44], long-chain: 14C–19C [41,45,46], and very-long-chain: ≥20C [17,47,48,49].

Short-chain FAs (SCFAs) are represented by two iFAs: 4:0 and 6:0 (Figure 2a). These are extremely rare in Embryophyta, and their median percentage in the FA composition of lipids trends towards zero. Medium-chain FAs (MCFAs) in Embryophyta are represented by 11 iFAs (Figure 2b). In vascular plants, they are mainly represented by saturated FAs (sFAs) such as 10:0, 11:0 and 12:0, while in mosses, the unsaturated FAs (uFAs) 12:1(7) and 13:1(8) are quite common. Ferns usually contain at least one MCFA in their lipid, while in other tribes, they are much less common. However, in gymnosperms, the median value of MCFAs is more than 1%, which is significantly higher than in others (Figure 2b). Long-chain FAs (LCFAs) in Embryophyta are the most diverse group and are represented by 67 iFAs (Figure 2c). They constitute the majority of all lipid FAs (median range from 75% to 95% in different tribes). Angiosperms exhibit the least diversity of LCFAs in a single species, while having the highest percentage content. In gymnosperms and spore plants, the diversity of LCFAs is higher, but the median content is lower. Of particular note is Bryophyta, which stands out for its greatest diversity of LCFAs, sometimes reaching 35 iFAs (Figure 2c). Very-long-chain FAs (VLCFAs) in Embryophyta are represented by 63 iFAs. The greatest diversity is observed in spore plants, slightly lower diversity is observed in gymnosperms, and the lowest diversity is observed in angiosperms. Quantitatively, their content is similar to species diversity by tribe, and in angiosperms, their median content does not exceed 4% (Figure 2d).

The acyl chains of lipid molecules play various roles in regulating the activity of membrane proteins and maintaining membrane function and integrity. The pleiotropic behavior of membrane properties shows a direct correlation with high levels of structural and the compositional complexity of the acyl chains of membrane lipids [50,51]. The length of the acyl chain is the main factor determining the thickness of the membrane [52]. It has been shown that changing the length of the acyl chain from 16 to 24 carbon atoms leads to a slight decrease in the average area per lipid and a clear linear increase in the thickness of the bilayer. Increasing the length of the acyl chain promotes interdigitation (interlinking of biological components that resembles the fingers of two hands being locked together) through the center of the bilayer, which is related to the molecules’ dynamics; this is because lateral diffusion (side-to-side movement of lipids within their respective leaflets of a bilayer membrane) rates decrease slightly as the acyl chain length increases [53].

The composition of membrane FAs determines the temperature of lipid phase transition (Tm) from the lamellar liquid crystalline (Lα) phase to the gel phase (Lβ). The longer the FA chains, the higher the transition temperature. For example, for the diacylphosphatidylethanolamine bilayer model, Tm changes from 30 °C to 90 °C if 12:0 is replaced with 22:0 [54].

Plants regulate the length of acyl chains to adapt to extreme temperatures. At low temperatures, chain shortening (e.g., an increase in the proportion of 16:3(7,10,13) in galactolipids) helps maintain membrane fluidity, preventing phase transition to Lβ [22]. On the contrary, at high temperatures, the elongation of acyl chains stabilizes membranes by strengthening hydrophobic interactions, so Arabidopsis mutants with defects in LCFA synthesis show increased sensitivity to heat stress [55].

In general, the interaction of FAs with lipid membranes is non-specific, weakly selective, and modulated by the physicochemical characteristics of the FA and the lipid bilayer under consideration. The incorporation of FAs into lipid membranes and their inversion across the bilayer occur almost instantaneously [56,57], and this, in turn, increases the curvature stress within the lipid bilayer due to the non-cylindrical geometry of FAs [58,59]. The local accumulation of FAs in the membrane will create small-scale inhomogeneities and defects in the membrane, possibly accompanied by toroidal pores and morphological changes in the membrane, all of which destabilize the membrane and reduce its permeability barrier [60,61]. While FAs with short acyl chains tend to destabilize the lipid bilayer, longer FAs can stabilize gel membranes, making them more rigid [62]. However, abnormal levels of FAs in some organisms, especially sFA, may accompany cellular dysfunction and damage [63,64,65]. FAs can exert their biological and pathological effects through receptor-mediated signaling, as well as through changes in the physical and mechanical properties of lipid bilayers [66]. The length of acyl chains modulates the activity of membrane-integrated proteins [67]. For example, plasma membrane ATPases require a specific lipid environment to function, so shortening the acyl chains of FAs disrupts the conformation of these ion transporters, reducing their activity [68]. Similarly, transport proteins such as aquaporins demonstrate the dependence of the water transport rate on the lipid bilayer thickness, regulated by chain length [69]. FAs play an important signaling role [70], and information about VLCFAs’ involvement in the response of plants to pathogen invasion is particularly noteworthy [71].

Despite the above-described features related to the length of FA chains in Embryophyta clades and the associated features of plant membrane functioning, it is noteworthy that an indicator such as the average chain length (ACL) [72] remains virtually unchanged. ACL consists of 18 carbon atoms and does not change significantly from tribe to tribe, with only a slight tendency towards chain shortening: mosses—18.0, ferns and gymnosperms—17.7, and angiosperms—17.5 (Appendix A). This suggests that ACL is crucial in the physiology of Embryophyta, and that rearrangements associated with changes in the chain length of some FAs must be compensated for by others.

## 5. Even and Odd Numbers of Carbon Atoms in the Acyl Chain

It should be noted that even-chain FAs (even-FAs) are characteristic of all higher plants. Despite the relatively high content of odd-FAs in certain species (for example, the maximum percentage of odd-FAs in angiosperms is 11.45 (sp. № 288), gymnosperms—35.7 (sp. № 344), ferns—18.3 (sp. № 454), mosses—9.3 (sp. № 531)), their median values are extremely low (0.27%, 1.8%, 0.8% and 0.45%, respectively, in the tribes) (Table A1 and Appendix A). In plants, FA synthesis primarily occurs in chloroplasts (plastids) and involves a series of enzymatic reactions in which fatty acid synthase (FAS) synthesizes FAs from acetyl-CoA, using malonyl-CoA as a carbon donor [73,74]. The process is cyclical, with two carbon atoms being added by FAS at a time until FA with 16 or 18 carbon atoms is formed [75]. Thus, even-FAs are obtained during de novo synthesis, but it is assumed that odd-FAs can be synthesized in a similar manner. [76], and de novo 15:0 may serve as the ancestor of this branch of the synthesis pathway [77]. It is also known that 18:0 can lose one carbon atom during α-oxidation, forming 17:0 [78,79,80,81]. As a result of the α-oxidation process occurring in peroxisomes, FAs are broken down into smaller molecules, such as acetyl-CoA, which can be used for the synthesis of vital chemicals or for energy production [82,83]. Unlike β-oxidation (which occurs in the mitochondria, during which two carbon atoms are cleaved from FAs molecules, ultimately forming acetyl-CoA and additionally NADH and FADH2 molecules, which are used for energy storage [84,85]), α-oxidation does not directly lead to energy storage [86]. However, there is evidence that α-oxidation is an important pathway for the metabolism of branched-chain FAs and may contribute to the incorporation of intermediate breakdown products into β-oxidation [87]. Nevertheless, it is evident that the appearance of odd-FAs indicates the initiation of energetically unfavorable FA catabolism, which can be observed under stressful conditions [17] and is not normal for Embryophyte.

## 6. The Double Bonds Number in the Fatty Acid Acyl Chain

Saturated FAs are those that do not contain double bonds in the acyl chain. The median sFA content is 32.5% in angiosperms, 32% in gymnosperms, 27% in ferns, and 22% in mosses (Figure 3; Appendix A).

Unsaturated FAs can contain varying numbers of double bonds in their acyl chain, and in Embryophyta they can range from one to six. The number of monoene FAs (mFAs) in the lipids of seed plants was practically equal (≈8%), and slightly higher in ferns and mosses (9–10%) (Figure 3; Appendix A). Diene FAs (dFAs) account for a slightly higher percentage of the total lipids in Embryophyta than mFAs, and there are no significant differences between tribes (Figure 3; Appendix A). Triene FAs (trFAs) are the most significant group of uFAs, with the highest levels found in gymnosperms and ferns (≈40%), angiosperms having 10% less, and mosses having another 10% less (Figure 3; Appendix A). The highest concentration of trFAs is found in the chloroplast membranes of all higher plants, suggesting that these FAs may be primarily necessary for photosynthesis [23,88]. The galactolipids of thylakoid membranes contain predominantly polyunsaturated C18 chains, which provide the flexibility necessary for the organization of photosystems II and I. Replacing 18:3(9,12,15) with 16:0 disrupts the structure of thylakoids and reduces the efficiency of photosynthesis [89,90]. At the same time, it is known that triple mutants of *fad3-2 fad7-2 fad8* Arabidopsis, which contain insignificant levels (less than 0.1%) of trFAs, did not differ from wild-type plants, and that their photosynthesis was practically unaffected [91]. However, these mutants were male sterile and did not produce seeds under normal conditions. The authors attribute this to the fact that trFAs are the precursors of oxylipins, which are signaling compounds that regulate the final maturation and release of pollen. At the same time, the exogenous application of jasmonic acid signaling molecules, formed in plants from 18:3(9,12,15) [92], or α-linolenic acid itself restored fertility [91], indicating the crucial role of trFAs in the life cycle of plants.

The most striking difference in uFA content is demonstrated by tetraene FAs (tFAs) (Figure 3; Appendix A). They are practically absent in angiosperms (median = 0). The presence of tFAs can be called an archaic feature of Embryophyta: the more evolutionarily the ancient tribe, the more tFAs these plants contain (gymnosperms—3.7%, ferns—5%, and mosses—9%). Pentaene FAs (pFAs) are quite rare in the composition of Embryophyta lipids and are completely uncharacteristic of seed plants. In ferns, their median value is 0.7%, with pFAs most commonly being found in mosses at 3%. (Figure 3; Appendix A). Hexaene FAs (hFAs) are not typical regarding the lipid composition of Embryophyta, occurring only in certain species of mosses (sp. № 528), ferns (sps. № 455 and № 457), and angiosperms, where the genus Centaurea should be singled out; in Centaurea, six species (sps. № 104, 105, 107, 111, 118, 121) contain gFAs, but these FAs are also found in species № 12, 15, 78, 163, 272, and 300 from various other families (Table A1, and Appendix A). Bryophytes often produce large amounts of very-long-chain polyunsaturated FAs (VLPUFAs), mainly including 20:4(5,8,11,14) and 20:5(5,8,11,14,17) [26]. The presence of VLPUFAs is characteristic of aquatic organisms such as algae [93,94,95], but it is rarely found in higher plants. This may indicate that the mosses did not completely lose their marine origins when they entered non-aquatic environments [96].

The presence and position of the double bond is just as important for the state of the membrane as the length of the acyl chain. Thus, on the model of a dioctadecenoyl phosphatidylcholine bilayer, it was shown that changing the position of the double bond from 4 to 9 or 13 in the carbon chains changes Tm from 20 °C to −20 °C or 0 °C, respectively [54]. Under physiological conditions, lipid bilayers are in the Lα state, in which lipids have considerable freedom of movement (they can rotate, swing, and diffuse laterally). When exposed to certain factors (e.g., temperature decrease, dehydration), FA lipid chains begin to pack into a crystal-like structure, forming a more ordered and stable Lβ, where the free movement of lipids is significantly restricted. Thus, the structure and geometry of the lipid acyl chains fundamentally influence the degree of packing within the bilayer. Any conformation of the acyl chain that hinders its tight packing will contribute to an increase in membrane fluidity. For example, compared to a straight sFA chain, the presence of an unsaturated *cis* bond within the chain will clearly limit the dense packing of the membrane [97]. The inclusion of a *cis* double bond causes the acyl chain to bend by 30°, which reduces the packing order in the lipid. This disorder is more pronounced when the *cis* bond is located in the middle of the acyl chains [98]. Thus, the uFA/sFA ratio is directly related to membrane fluidity [99,100,101].

The unsaturation index (UI) is commonly used as an integral indicator that most clearly demonstrates the number of double bonds in the FA composition [102,103]. This indicator illustrates the gradual decrease in unsaturation during the evolutionary process, i.e., from mosses to angiosperms (Figure 4). Moreover, the highest UI values in mosses and ferns are achieved due to tFAS and pFAs, while in gymnosperms, PUFA data are reduced; however, the UI remains quite high due to the high content of trFAs. Thus, the decrease in the amount of VLCFAs (Figure 2d) is accompanied by a significant decrease in the unsaturation of FAs in the lipid composition from mosses to angiosperms. This is quite logical, since it is known that the membrane fluidity will increase both when the FA chain of the membrane is shortened [104] and when the *iso* bonds in FA are increased [105].

With their emergence onto land [106], plants were constantly confronted with changes in climate, with hot periods alternating with prolonged ice ages [107]; it is clear that these stressful conditions promoted the evolution of Embryophyta [108,109]. It can be assumed that repeated periods of decline in the average temperature of the planet [110] contributed to the inclusion of a larger number of shorter-chain FAs in the composition of lipids, since it is known that they help maintain membrane fluidity under cold stress [22]. As one of the compensatory mechanisms for maintaining the physiological parameters of membrane viscosity, plants have been forced to reduce the unsaturation of their membranes during evolution, since these indicators, when changing in one direction, lead to opposite effects on the value of Tm.

It is known that the FA profile is associated with the thermotolerance of plants in different climatic zones [111]. Numerous individual cases of increased FA unsaturation have been identified; this occurs during salinization, drought and UV radiation [112]. The change in membrane fluidity, mediated by changes in unsaturated FA levels, is a function provided in part by the regulated activity of FA desaturases. Various studies have been devoted to the evolution of these enzymes [10,11,113,114,115], so we will not dwell on them separately; however, we cannot fail to mention the important regulatory role of desaturases. Thus, the regulation of membrane fluidity maintains an environment suitable for the functioning of critically important integral proteins during stress. For example, the modulation of 18:1(9) content is central to the normal expression of defense responses to pathogens in Arabidopsis, and the control of 18:1(9) and 18:2(9,12) levels is involved in the regulation of seed development and mycotoxin production against *Aspergillus* spp. [116].

## 7. Conclusions

Thus, plants adapt the composition of their membrane FAs to specific conditions, which leads to diversity in FAs and their composition among different species. However, all these changes, consisting of the inclusion of certain FAs in the composition of lipids, are compensated for either by the inclusion/exclusion of other FAs or by a change in their level of unsaturation. This is necessary to maintain certain physiological parameters, such as ACL, and is observed throughout the development of Embryophyta; this ensures optimal Tm values that are directly related to the conductive and protective functions of the lipid bilayer, which constitutes the fundamental physiological functions of the plant organism.

It is also important to note that lipids containing essential FAs are an indispensable component of human nutrition. Therefore, it is necessary to expand research into the FA profile of plants from different taxonomic groups. This will increase breeding options when creating mutants with specified characteristics, considering the physiological capabilities and compensatory functions of the plant organism. At present, in order to comprehensively address this issue, it is necessary to create complete and publicly accessible databases of chromatographic profiles (with mass spectrometric confirmation) for the lipidomes of representatives of all systematic groups of higher plants, since most of the data currently available is scattered and often incomplete.

## Figures and Tables

**Figure 1 ijms-26-09424-f001:**
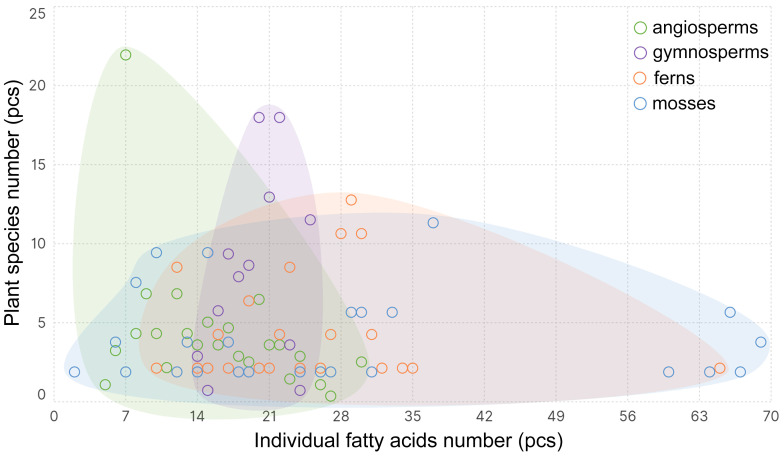
The frequency with which a certain number of individual fatty acids occur in different Embryophyta clades: mosses, ferns, gymnosperms, and angiosperms.

**Figure 2 ijms-26-09424-f002:**
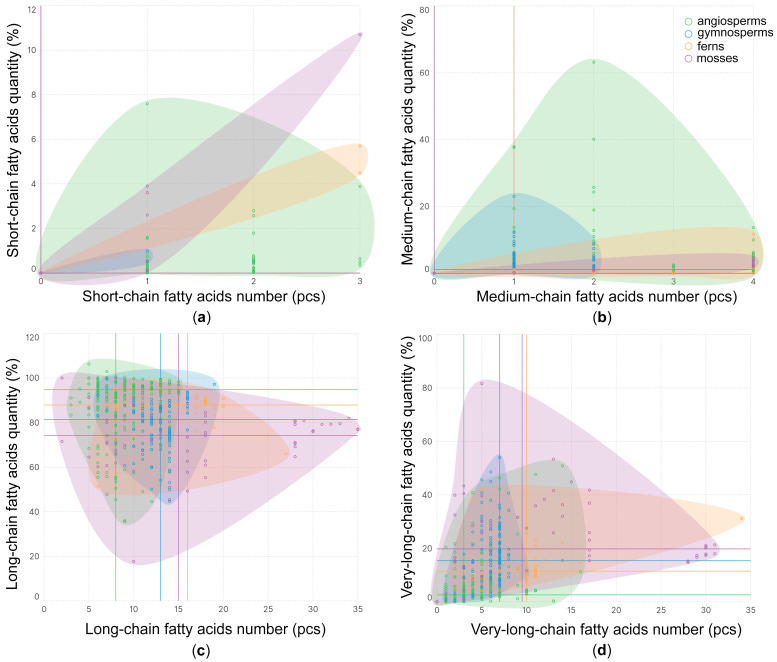
The number of short-chain (**a**), medium-chain (**b**), long-chain (**c**), and very-long-chain (**d**) fatty acids in different species of Embryophyta clades: mosses, ferns, gymnosperms, and angiosperms. The lines corresponding to the legend colors indicate the median value of the qualitative and quantitative composition of lipid fatty acids with a specific acyl chain length according to the clades.

**Figure 3 ijms-26-09424-f003:**
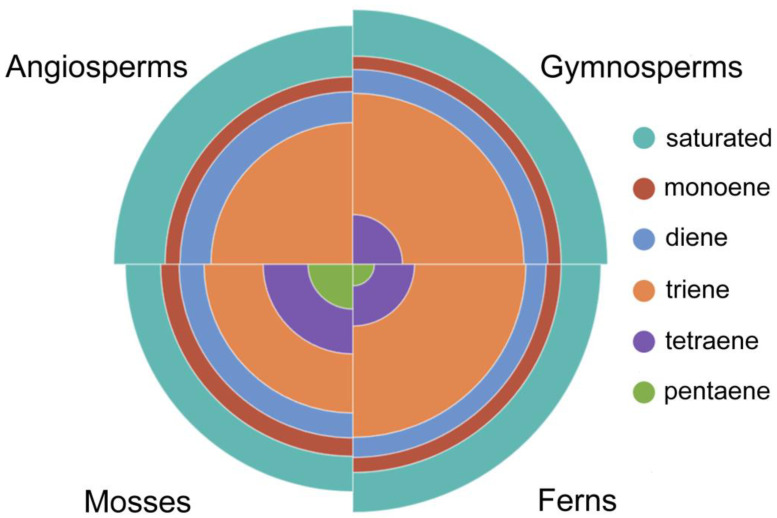
Median distribution profile of saturated and unsaturated (monoene, diene, triene, tetraene, and pentaene) lipid fatty acids in Embryophyta clades: mosses, ferns, gymnosperms, and angiosperms.

**Figure 4 ijms-26-09424-f004:**
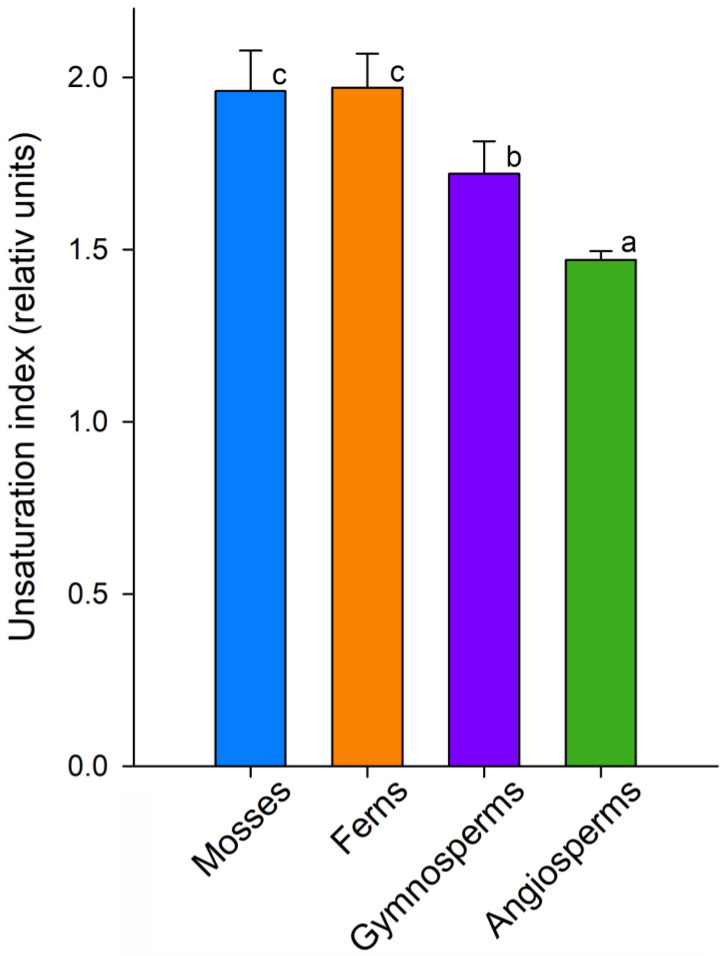
Unsaturation index of various Embryophyta clades: mosses, ferns, gymnosperms, and angiosperms. Values assigned with different letters indicate a significant difference between the medians (one-way ANOVA followed by post hoc Tukey’s NHSD test, *p* < 0.05).

## Data Availability

All the data are provided in the review.

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
