# Peer review of "Phylogenetic Aspects of Higher Plant Lipid Fatty Acid Profile"

_ijms, 2025, doi:10.3390/ijms26199424_

Round 1

Reviewer 1 Report

Comments and Suggestions for Authors

This paper compiles information on fatty acid accumulation in 557 species of land plants. However, the data presented have not been sufficiently validated, and the reviewer has judged the manuscript unsuitable for publication in JIMS in the present version.

The primary concern is that the fatty acid profile data for the 557 species were obtained under varying experimental conditions. As a result, the comparative analysis of fatty acid types and compositions across species is of questionable significance. To address this issue, the reviewer recommends that information on plant growth conditions, developmental stages, sampled organs or tissues, and the methods used for fatty acid analysis be compiled from each source paper. This would allow for better standardization and more meaningful comparisons.

Furthermore, the reviewer was unable to verify the accuracy of Tables S1 and S2 or Figures 1–4, as the specific source references for each dataset are not cited within the tables.

Figure 1:

This figure is potentially misleading. The meaning of the highlighted (bright) areas should be explicitly clarified. For instance, the plot at X=7 and Y=22 likely correspond to Nicotiana species, which may create the false impression that angiosperms contain fewer fatty acid species. Likewise, the few plots at around X=56–70, which represent ferns and mosses, may erroneously suggest that these groups have more diverse fatty acid profiles.

Table S1:

The fatty acid notations used are incorrect and should be revised. According to the LIPID MAPS classification system, for example, alpha-linolenic acid should be denoted as FA 18:3 (9,12,15) or FA 18:3 (9Z,12Z,15Z).

Table S2:

Each column header should be clearly defined. Abbreviations such as "ACL" and "UI" are not explained and lack context, making interpretation difficult.

Author Response

Dear Reviewer, Thank you for your detailed review of our manuscript. We have taken into account all your comments and thank you for helping us improve our work. Below we provide answers to your comments point by point.

The primary concern is that the fatty acid profile data for the 557 species were obtained under varying experimental conditions. As a result, the comparative analysis of fatty acid types and compositions across species is of questionable significance. To address this issue, the reviewer recommends that information on plant growth conditions, developmental stages, sampled organs or tissues, and the methods used for fatty acid analysis be compiled from each source paper. This would allow for better standardization and more meaningful comparisons.

Thank you for drawing attention to the absence of this information in the article, we naturally took all these factors into account when analyzing the material, and added the corresponding paragraph to the manuscript.

Furthermore, the reviewer was unable to verify the accuracy of Tables S1 and S2 or Figures 1–4, as the specific source references for each dataset are not cited within the tables.

Thank you very much for your comment. The numbering of species in all tables is uniform, and the sources of literature are given in Table A1. For the convenience of readers, we have also duplicated the references to literature in Tables S1 and S2.

Figure 1:

This figure is potentially misleading. The meaning of the highlighted (bright) areas should be explicitly clarified. For instance, the plot at X=7 and Y=22 likely correspond to Nicotiana species, which may create the false impression that angiosperms contain fewer fatty acid species. Likewise, the few plots at around X=56–70, which represent ferns and mosses, may erroneously suggest that these groups have more diverse fatty acid profiles.

The confusion probably arose from an unsuccessful caption to the figure. This figure does not demonstrate the number of acids in a specific species, but shows how many species in a given clade contain such a specific amount of FA. Therefore, there is no error, especially since we use median values for analysis and individual outliers in the composition are not taken into account. We have rewritten the caption to the figure so that its information becomes clearer.

Table S1:

The fatty acid notations used are incorrect and should be revised. According to the LIPID MAPS classification system, for example, alpha-linolenic acid should be denoted as FA 18:3 (9,12,15) or FA 18:3 (9Z,12Z,15Z).

We thank the reviewer for the comment on the classification of the FAs we have chosen; we have made the appropriate corrections to the entire manuscript.

Table S2:

Each column header should be clearly defined. Abbreviations such as "ACL" and "UI" are not explained and lack context, making interpretation difficult.

Thank you for your attentive attention to our manuscript, it has been corrected.

Reviewer 2 Report

Comments and Suggestions for Authors

Review report

Review: Phylogenetic aspects of higher plant lipid fatty acid profile

General Comments:

The review is well-structured and scientifically rigorous, progressing logically from background to detailed analysis and conclusion.

The writing style is formal and precise but highly technical, which limits accessibility for readers not expert in lipid biochemistry or plant physiology.

Using more illustrative diagrams, simplified explanatory sidebars, or summary boxes could break down complex information and improve readability.

Minor editorial revisions could reduce sentence length and improve flow without compromising scientific accuracy

  1. Title and abstract:

The title is precise and immediately indicates the phylogenetic perspective on plant lipid fatty acids, which sets clear expectations for the reader.

The abstract successfully conveys the scope and significance of the review but is technically dense. For example, terms like "temperature of phase transition" and "unsaturation index" are not explained, which could confuse readers unfamiliar with lipid biochemistry.

The abstract mentions the number of species covered L550) but does not highlight key patterns or discoveries such as the trend of decreasing fatty acid diversity from mosses to angiosperms, which would make it more informative and impactful.

  1. Introduction

The introduction provides strong context by contrasting the extensive knowledge of DNA and proteins with the gap in understanding lipid fatty acid evolution.

It clearly states the novelty and rationale for this review but could be more engaging by briefly mentioning the physiological roles of fatty acids, such as membrane structure and adaptability to environmental stress.

The large number of citations early on might overwhelm readers. Selective referencing, focusing only on the most critical background studies, would improve readability.

  1. Diversity of Fatty Acids

This section effectively catalogs the fatty acid diversity among plant groups, showing a clear pattern of reduction in fatty acid types through evolution.

The authors propose logical adaptive reasons, such as loss of complexity due to improved water management in vascular plants, and specialization of metabolic pathways in angiosperms.

However, many evolutionary explanations are speculative without direct experimental validation within the review. Marking these as hypotheses and suggesting experimental avenues for future work would clarify their status.

The presentation would benefit from summary tables or graphical data, such as bar charts or heat maps, which would help readers visualize the trends across species groups.

  1. Major Fatty Acids

The review identifies a core set of major fatty acids common to higher plants and appropriately notes unusual profiles like those in Nicotiana.

It links biochemical differences to ecological and functional roles, for example, the role of sugar polyesters in plant defense, enriching the biological relevance.

The section sometimes conflates fatty acid composition data with metabolic function without clearly separating these topics, potentially confusing the reader.

Additional discussion on evolutionary pressures shaping the conservation or divergence of these major fatty acids could deepen the readersʼ understanding.

  1. Acyl Chain Length

This section thoroughly reviews fatty acid chain length classes, their diversity, and their influence on membrane properties like thickness and fluidity.

Integration of biophysical data with evolutionary observations is a strength, showing how chain length relates to membrane stability and plant adaptation.

The detailed explanation of α- and β-oxidation pathways, while informative, may distract from the main focus. Condensing this discussion or moving it to a dedicated subsection would improve flow.

Complex terms such as "interdigitation" and "lateral diffusion" are used without definition; quick explanations or glossary inclusion would help non-specialist readers.

  1. Unsaturation and Double Bonds

The review gives an excellent quantitative comparison of saturation levels and unsaturation degrees across plant groups with meaningful biological context such as membrane fluidity and temperature adaptation.

The evolutionary trend showing decreased unsaturation in more derived plant groups is well presented.

However, the depth of technical detail could be overwhelming. Including diagrams like a schematic of fatty acid unsaturation or a graphical abstract summarizing these data would aid comprehension.

Some apparent contradictions, such as the essential nature of triene fatty acids for photosynthesis versus mutant plants with very low amounts having normal photosynthesis but reduced fertility, are mentioned but not fully reconciled. Proposing hypotheses to explain these findings or discussing the different roles of triene fatty acids would strengthen the argument.

  1. Conclusions

The conclusions summarize key findings effectively, emphasizing the compensatory mechanisms plants use to maintain membrane stability through fatty acid composition and unsaturation.

The mention of practical significance, such as human nutrition and plant breeding, highlights broader relevance.

The section could be improved by clearly stating specific unresolved questions and suggesting directions for future research to guide readers interested in further study.

Author Response

Dear Reviewer, thank you for your careful work with our manuscript. Your comments helped to significantly improve our paper, all your comments were taken into account, and below we provide detailed answers to them point by point.

General Comments:

The review is well-structured and scientifically rigorous, progressing logically from background to detailed analysis and conclusion.

The writing style is formal and precise but highly technical, which limits accessibility for readers not expert in lipid biochemistry or plant physiology.

Using more illustrative diagrams, simplified explanatory sidebars, or summary boxes could break down complex information and improve readability.

Minor editorial revisions could reduce sentence length and improve flow without compromising scientific accuracy

We clarified some complex concepts in the manuscript, supplemented the paper with an additional figure for clarity, and also improved the English language using the MDPI service.

1. Title and abstract:

The title is precise and immediately indicates the phylogenetic perspective on plant lipid fatty acids, which sets clear expectations for the reader.

The abstract successfully conveys the scope and significance of the review but is technically dense. For example, terms like "temperature of phase transition" and "unsaturation index" are not explained, which could confuse readers unfamiliar with lipid biochemistry.

The abstract mentions the number of species covered L550) but does not highlight key patterns or discoveries such as the trend of decreasing fatty acid diversity from mosses to angiosperms, which would make it more informative and impactful.

Thank you for your valuable comments. The Abstract has been revised taking them into account.

2. Introduction

The introduction provides strong context by contrasting the extensive knowledge of DNA and proteins with the gap in understanding lipid fatty acid evolution.

It clearly states the novelty and rationale for this review but could be more engaging by briefly mentioning the physiological roles of fatty acids, such as membrane structure and adaptability to environmental stress.

The large number of citations early on might overwhelm readers. Selective referencing, focusing only on the most critical background studies, would improve readability.

Thank you very much for your careful reading of the manuscript. We have tried to make the text easier to read and also expanded this section.

3. Diversity of Fatty Acids

This section effectively catalogs the fatty acid diversity among plant groups, showing a clear pattern of reduction in fatty acid types through evolution.

The authors propose logical adaptive reasons, such as loss of complexity due to improved water management in vascular plants, and specialization of metabolic pathways in angiosperms.

However, many evolutionary explanations are speculative without direct experimental validation within the review. Marking these as hypotheses and suggesting experimental avenues for future work would clarify their status.

The presentation would benefit from summary tables or graphical data, such as bar charts or heat maps, which would help readers visualize the trends across species groups.

Thank you very much for the high evaluation of our work and the clear indication of the problem areas. We have included information that these are hypotheses and also prepared heat maps (Figure S1) for better visualization of the material.

4. Major Fatty Acids

The review identifies a core set of major fatty acids common to higher plants and appropriately notes unusual profiles like those in Nicotiana.

It links biochemical differences to ecological and functional roles, for example, the role of sugar polyesters in plant defense, enriching the biological relevance.

The section sometimes conflates fatty acid composition data with metabolic function without clearly separating these topics, potentially confusing the reader.

Additional discussion on evolutionary pressures shaping the conservation or divergence of these major fatty acids could deepen the readersʼ understanding.

We have adjusted the text according to your suggestions and expanded the discussion of the development of evolutionary factors of these phenomena.

5. Acyl Chain Length

This section thoroughly reviews fatty acid chain length classes, their diversity, and their influence on membrane properties like thickness and fluidity.

Integration of biophysical data with evolutionary observations is a strength, showing how chain length relates to membrane stability and plant adaptation.

The detailed explanation of α- and β-oxidation pathways, while informative, may distract from the main focus. Condensing this discussion or moving it to a dedicated subsection would improve flow.

Complex terms such as "interdigitation" and "lateral diffusion" are used without definition; quick explanations or glossary inclusion would help non-specialist readers.

Thank you for your suggestions, we have highlighted an additional section in the manuscript and clarified terms that are difficult for non-specialist readers.

6. Unsaturation and Double Bonds

The review gives an excellent quantitative comparison of saturation levels and unsaturation degrees across plant groups with meaningful biological context such as membrane fluidity and temperature adaptation.

The evolutionary trend showing decreased unsaturation in more derived plant groups is well presented.

However, the depth of technical detail could be overwhelming. Including diagrams like a schematic of fatty acid unsaturation or a graphical abstract summarizing these data would aid comprehension.

Some apparent contradictions, such as the essential nature of triene fatty acids for photosynthesis versus mutant plants with very low amounts having normal photosynthesis but reduced fertility, are mentioned but not fully reconciled. Proposing hypotheses to explain these findings or discussing the different roles of triene fatty acids would strengthen the argument.

Thank you for your comment. This section contains graphical materials to summarize the data, facilitating understanding of the material. The discussion of the role of triene FAs has been strengthened.

7. Conclusions

The conclusions summarize key findings effectively, emphasizing the compensatory mechanisms plants use to maintain membrane stability through fatty acid composition and unsaturation.

The mention of practical significance, such as human nutrition and plant breeding, highlights broader relevance.

The section could be improved by clearly stating specific unresolved questions and suggesting directions for future research to guide readers interested in further study.

We strengthened the conclusion with information on the role of FA in human nutrition and plant breeding. A vector was proposed for the development of the study and application of information on the FA composition of higher plants.